# miR-450-5p and miR-202-5p Synergistically Regulate Follicle Development in Black Goat

**DOI:** 10.3390/ijms24010401

**Published:** 2022-12-26

**Authors:** Guanghang Feng, Jie Liu, Zitao Lu, Yaokun Li, Ming Deng, Guangbin Liu, Baoli Sun, Yongqing Guo, Xian Zou, Dewu Liu

**Affiliations:** 1Herbivore Laboratory, College of Animal Science, South China Agricultural University, Guangzhou 510642, China; 2State Key Laboratory of Livestock and Poultry Breeding, Guangdong Key Laboratory of Animal Breeding and Nutrition, Institute of Animal Science, Guangdong Academy of Agricultural Sciences, Guangzhou 510640, China; 3National Joint Engineering Research Center, South China Agricultural University, Guangzhou 510642, China; 4Guangdong Key Laboratory of Agricultural Animal Genomics and Molecular Breeding, South China Agricultural University, Guangzhou 510642, China; 5Collaborative Innovation Center for Healthy Sheep Breeding and Zoonoses Prevention and Control, Shihezi University, Shihezi 832000, China

**Keywords:** goat, follicle development, granulosa cells, MiR-202-5p, MiR-450-5p

## Abstract

Follicle maturation is a complex biological process governed by numerous factors, and researchers have observed follicle development by studying the proliferation and apoptosis of follicular granulosa cells (GCs). However, the regulatory mechanisms of GCs proliferation and death during follicle development are largely unknown. To investigate the regulatory mechanisms of lncRNAs, mRNAs, and microRNAs, RNA sequencing (RNA-seq) and small RNA-seq were performed on large (>10 mm) and small follicles (<3 mm) of Leizhou black goat during estrus. We discovered two microRNAs, miR-450-5p and miR-202-5p, which can target GCs in goats and may be involved in follicle maturation, and the effects of miR-450-5p and miR-202-5p on ovarian granulosa cell lines were investigated (KGN). Using cell counting kit-8 (CCK-8) assays, 5-Ethynyl-2’-deoxyuridine (EdU) assay and flow cytometry, miR-202-5p overexpression could suppress the proliferation and induce apoptosis of GCs, whereas miR-450-5p overexpression induced the opposite effects. The dual-luciferase reporter assay confirmed that miR-450-5p could directly target the BMF gene (a BCL2 modifying factor), and miR-202-5p targeted the *BCL2* gene. A considerable rise in phosphorylated Akt (p-AKT) protein was observed following the downregulation of *BMF* by miR-450-5p mimics. After *BMF* gene RNAi therapy, a notable elevation in p-AKT was detected. Mimics of miR-202-5p inhibited BCL2 protein expression, significantly decreasing p-AMPK protein expression. These results imply that during the follicular development in black goats, the miR-450-5p-*BMF* axis favored GC proliferation on a wide scale, while the miR-202-5p-*BCL2* axis triggered GC apoptosis.

## 1. Introduction

Mammalian follicle development is a complex and carefully-regulated process, ranging from primordial follicle activation to successful ovulation and including follicular atresia and degeneration [1]. Granulosa cells (GCs) offer a stable internal environment for oocyte development as a component of the follicle t [2]. Furthermore, GCs can create estrogen and produce follicle-stimulating hormone (FSH) and luteinizing hormone (LH) receptors to regulate follicle development [3,4]. Most previous studies employed the transcriptome of GCs as a starting point to examine the features of follicle development, particularly the oocyte maturation [5,6]. Nevertheless, because the follicular development process is dynamic, the varying biases of each study have resulted in a less exhaustive analysis of GCs.

MicroRNAs (miRNAs), the non-coding single-stranded RNA molecules, are encoded by endogenous genes and participate in the post-transcriptional regulation of gene expression in plants and animals [7]. In recent years, studies of miRNAs as entry points to animal reproductive performance have accelerated our understanding of the genetic mechanisms underlying follicle development. Abu et al. emphasized the importance of miRNAs in various aspects of mammalian reproduction, such as germ cell biogenesis, reproductive organ function, and early embryonic development [8]. Let-7e inhibits the p21 signaling pathway in polycystic ovary syndrome to regulate human GC proliferation and autophagy without causing hyperandrogenemia [9]. MiR-202 knockout by CRISPR/Cas9 gene editing tools in medaka can result in reduced or no spawning; additionally, miR-202-5p has been shown to target *TGFβR2* to promote apoptosis in velvet goat GCs [10,11]. Not only that, but the loss of miR-7a was found to cause hypogonadism and sterility in mice, and the reproductive performance of chi-miR-101-3p in goat ovaries was significantly different [12,13].

Due to the complexity of follicular development, the specific regulatory mechanisms of miRNAs remain unknown, even though they are widely involved in numerous mammalian developmental processes. In this experiment, we respectively analyzed the large and small follicles of Leizhou goats using RNA sequencing and small-RNA sequencing, combined with the laboratory sequencing data of Chuanzhong goats [14]. Based on the Gene Ontology (GO) and Kyoto Encyclopedia of Genes and Genomes (KEGG) databases, the miRNAs with significant differences were screened. Specifically, miR-450-5p and miR-202-5p were chosen in this study. In order to provide a reference for the study of mammalian follicle development, we investigated their function in ovarian granulosa cell lines (KGN), followed by identifying their target genes and pathways in the apoptotic pathway

## 2. Results

### 2.1. Raw Data Processing, Filtering and Quality Assessment

Each goat had one-to-two large follicles (d > 10 mm) and eight-to-fifteen small follicles (d < 3 mm) isolated. Seven Leizhou goats yielded eight large follicles and five small follicular pools. Large follicles with few or no blood vessels on the surface were deemed atretic and therefore not used for RNA-seq. In the end, only five large follicles were sequenced. For the preparation of large follicles (one follicle per sample) and small follicles, RNA-seq libraries were utilized (eight-to-fifteen follicles per sample). The original reads of the large and small follicular groups of Leizhou goats were subsequently analyzed for quality control. Due to the presence of adapters and low-quality reads, which may significantly interfere with subsequent information analysis, the sequencing data must be further filtered. The results demonstrated that the Q30 values of each sample exceeded 91% (Table 1), and more than 82% (Appendix A) of the total reads could be mapped to the reference genome with a high degree of similarity, indicating that the sequencing data were of high quality and suitable for further analysis.

### 2.2. Differentially Expressed mRNAs, lncRNAs and miRNAs

The criteria of expression difference fold |log2FoldChange| > 1 and significant *p* value < 0.05 were used to identify the differentially expressed genes. Up to 3000 DEmRNAs were detected in the LF (*n* = 5) and SF (*n* = 5) groups, with 2144 being upregulated in the SF group (Figure 1A). Cluster analysis of the differential expression genes separated the samples into two expected categories (Figure 1B). Furthermore, up to 261 annotated DElncRNAs were found in the SF group, and 28 were down-regulated (Figure 1C). DElncRNA clustering analysis revealed good integration between the two sample groups (Figure 1D). Finally, 44 candidate miRNAs were found to be differentially expressed, with 24 miRNAs being down-regulated in the SF group (Figure 1E). As expected, the clustering analysis results explained the significant differences between the two sample groups (Figure 1F). In total, 3000 DEmRNAs, 261 DElncRNAs, and 44 DEmiRNAs were detected in small follicles versus large follicles, with Table 2 listing the top ten DEmRNAs, DElncRNAs, and DEmiRNAs.

### 2.3. Functional Analysis of DEmRNAs and Target Genes of DElncRNAs and DEmiRNAs

Between the two groups, up to 3000 DEmRNAs were found, of which 2144 were upregulated, and 856 were downregulated in small follicles. GO analysis of DEmRNAs revealed the enrichment of multicellular organism processes, transporter activity, and plasma membrane components. According to KEGG pathway analysis, the DEmRNAs were enriched in the immune-system pathway, nutrient digestion and metabolism, endocrine processes, and lipid synthesis and metabolism. Particularly, strong enrichment was observed in the hormone-related signaling pathways, including insulin secretion, ovarian steroidogenesis, steroid hormone biosynthesis, thyroid hormone synthesis, and insulin signaling pathways, as well as the signal transduction-related pathways, such as the ABC transporter pathway and cell adhesion molecules (CAMs) (Figure 2B).

Through comparing with the reference genome, 23 of the 261 annotated DElncRNAs were identified as novel lncRNAs. According to biological process (BP), cell component (CC), and molecular function (MF), a GO analysis of the predicted lncRNA target genes was performed. The majority of functional terms were enriched in cell development and differentiation, nucleic acid binding transcription activity, and postsynaptic membrane components (Figure 2C). As determined by KEGG Pathway analysis, the immune system processes (Th17 cell differentiation), p53 signaling pathway, and cell adhesion molecules were involved in the key pathways implicated by the targeted genes (Figure 2D).

This study identified a total of 44 candidate DEmiRNAs, of which 24 were down-regulated in small follicles. A pathway enrichment analysis of DEmiRNAs was conducted, and 40 KEGG pathways were assigned to the target genes. The TGF-beta signaling pathway, cell-adhesion molecules, MAPK signaling pathway, and steroid hormone biosynthesis were significantly enriched and highly associated with follicular development (Figure 2F). Furthermore, GO analysis revealed that DEmiRNA target genes are involved in cellular development and nucleic acid binding transcription activity (Figure 2E).

### 2.4. Real-Time Fluorescence Quantitative Validation

mRNAs, lncRNAs, and miRNAs of interest were selected for validation by the quantitative real-time PCR based on the RNA-seq results. The expression of the ten selected mRNAs (Figure 3A,B), five lncRNAs (Figure 3C,D), and two miRNAs (Figure 3E,F) in goat follicles was consistent with the sequencing results, indicating that the sequencing results were stable and dependable. Primer information is listed in Appendix A.

### 2.5. miR-450-5p and miR-202-5p Are Upregulated in GCs Expression

GCs exhibit apoptosis as they rapidly grow during follicle development. Follicle size grows with the GCs proliferation, while GCs apoptotic allow for the successful expulsion of oocyte from the follicle. We chose miR-450-5p and miR-202-5p for further research because of their significant enrichment in the healthy and large follicles of both Leizhou and Chuanzhong goats during ertrus [14]. Herein, we showed that miR-450-5p and miR-202-5p are specifically expressed in GCs but not in theca cells of large follicles in goat ovaries (Figure 4A). Several previous studies suggest that miR-202-5p accumulated in extracellular vesicles is functional in the GCs during follicular growth and may regulate oocyte maturation in goats [11]. In contrast, miR-450-5p is a novel miRNA highly expressed in large follicular granulosa cells, and no study has reported its role in follicular development.

### 2.6. miR-450-5p Stimulates the Proliferation of GCs

To investigate the effect of miR-450-5p on granulosa cells, miR-450-5p mimics were transfected into KGN. The expression of miR-450-5p was increased 12,000-fold (Appendix A). At 72 h, the Cell Counting Kit 8 revealed that the number of proliferating cells with miR-450-5p overexpression was significantly greater than in the control group (Figure 4B). Using a flow cytometry assay, it was discovered that the apoptosis rate in the experimental group was 4.22%, lower than that of the control group (Figure 4C,D). In addition, the EdU proliferation assay revealed that the cell proliferation rate of the experimental group was significantly greater than that of the control group (Figure 4E,F). These results indicate that miR-450-5p can promote the proliferation of follicular GCs for the first time.

### 2.7. miR-450-5p Activates p-AKT Expression by Targeting BMF

In this study, miR-450-5p was found to be highly expressed in large follicles (Figure 3E,F). MiRNAs typically target genes 3’ UTRs and exert regulatory effects. The miR-450-5p target genes were predicted by miRDB software (Figure 5A), and three potential target genes *(BMF*, *TNRC6C,* and *RS1)* were chosen for further investigation due to their significant upregulation in small follicles (Table 3). By qRT-PCR, it was determined that the *BMF* gene was highly expressed in small follicles (Figure 5B), and overexpression of miR-450-5p in KGN cells could inhibit *BMF* gene expression (Figure 5C). *BMF* (*BCL-2* modifying factor) is a classical gene that regulates the apoptotic pathway and has been shown to promote the proliferation of goat follicular GCs, which play a crucial role in follicular development in animals [15]. Therefore, we conducted dual luciferase validation experiments utilizing the constructed wild-type (WT) and mutant plasmids (MUT) in addition to the original null plasmid (PGLOmir). The fluorescence of the MUT and PGLOmir groups did not differ significantly from their respective controls, but of the WT group was suppressed upon miR-450-5p overexpression (Figure 5D,E). Notably, overexpression of miR-450-5p significantly inhibited BMF protein expression (Figure 5F), suggesting that *BMF* is a miR-450-5p target gene. AKT protein expression was not significantly different after overexpression of miR-450-5p compared to the control group, whereas p-AKT protein expression was significantly higher (*p* < 0.05) (Figure 5F,G). We designed siRNA to knock down the *BMF* gene, and qRT-PCR and Western blot analyses revealed that the *BMF* gene and protein expression was drastically reduced (Figure 5H,I). Notably, *BMF* knockdown did not significantly alter AKT protein expression, whereas p-AKT protein expression was significantly elevated compared with controls (Figure 5I,J). These results suggest that miR-450-5p promotes GCs proliferation by targeting *BMF* and influencing the PI3K/Akt signaling pathway.

### 2.8. miR-202-5p Induces the Apoptosis of Follicular GCs

KGN cells were transfected with miR-202-5p mimics (Appendix A), and a flow cytometry assay revealed that the apoptosis rate rose by 2% in the miR-202-5p group relative to the NC group (Figure 6A,B), suggesting that the number of apoptotic cells was more than that of the model NC group. In addition, the proliferating cells’ OD at 48 h and 72 h were significantly lower in the model group than in the model NC group, as measured by Cell Counting Kit 8 (Figure 6C). Furthermore, the EDU cell proliferation assay demonstrated that miR-202-5p could inhibit the proliferation of GCs (Figure 6D,E), indicating that miR-202-5p could promote the apoptosis of follicular granulosa cells.

### 2.9. miR-202-5p Binds BCL2 to Trigger Apoptosis in GCs via Blocking the AMPK Signaling Pathway

It has been discovered that miR-202-5p, substantially expressed in big follicles, can regulate follicular GC development in humans by targeting *TGFβR2* and *BCL2* [11]. However, the genetic diversity of animals and the existence of more miRNA target genes complicate matters. We matched the differential mRNAs strongly expressed in tiny follicles with the anticipated miR-202-5p target genes and discovered that *BCL2* might be a possible target gene in humans and goats (Figure 7A). *BCL2* was extensively confirmed as an anti-apoptotic gene, and quantitative data demonstrated that miR-202-5p could inhibit the expression of *BCL2* gene and protein (Figure 7B,E). Dual-luciferase validation demonstrated that the fluorescence of the MUT and PGLOmir groups was unaffected by miR-202-5p overexpression, but the fluorescence of the WT group was decreased (Figure 7C,D). These data imply that *BCL2* is a miR-202-5p target gene. In this work, a substantial number of target genes of differential miRNAs were considerably enriched in the AMPK signaling pathway, which participates in regulating cell growth and development. Consequently, overexpression of miR-202-5p caused no significant difference in AMPK protein expression compared to the control group, although p-AMPK protein expression was considerably decreased (Figure 7E,F). Therefore, miR-202-5p can target *BCL2* and disrupt the AMPK signaling pathway to influence follicle growth. In conclusion, we discovered that miR-450-5p and miR-202-5p dynamically regulate GCs growth. GCs proliferation enables bigger follicle growth, and GCs death facilitates the expulsion of follicle cells through the GCs layer (Figure 8).

## 3. Discussion

This study employed next-generation sequencing to offer a quantitative and exhaustive investigation of the coding and non-coding transcriptome in the large and small follicles of Leizhou goats. These results indicated significant changes in the expression patterns of mRNA, lncRNA, and miRNA that may be implicated in the regulatory mechanisms controlling follicular maturation and development.

RNA-seq found a total of 3000 DEmRNAs, 261 DElncRNAs, and 44 DEmiRNAs. Many of the mRNAs identified from our sequencing results have been implicated in regulating follicle growth in other species. For example, inhibin A, encoded by *INHBA* and *INHA*, regulates estradiol secretion, increases follicle sensitivity to FSH, and prevents follicular atresia throughout the dominant follicle growth stage [16,17]. The four major genes, *LHCGR, CYP19A, CYP11A1,* and *HSD3B*, controlled the key node in the steroid hormone synthesis pathway [18,19]. LncRNA functions as a sponge to absorb miRNAs to regulate genes, and in the endometrial epithelial cells of dairy goat, LncRNA882 modulates leukemia inhibitory factor (LIF) via sponge adsorption of miR-15b [20]. A large variety of lncRNAs have been linked to follicular formation, and their roles in GC proliferation (ENSCHIT00000001444, MSTRG.4906.1, ENSCHIT00000005590) will be studied in the following study.

Similar to previous research on follicular development, KEGG pathway enrichment demonstrated that steroid hormone production (estradiol, testosterone, progesterone, etc.) was considerably enriched in DEmRNAs [21,22]. It inhibited apoptosis in the preantral and early antral follicular granulosa cells and preserved the arrest of oocyte meiosis [23]. In addition, oxidative phosphorylation, cAMP signaling route, and chemokine signaling pathway were implicated in our data, thereby enhancing our understanding of follicular development [24]. And the future study should focus on the upstream regulators of the follicular development pathway. The majority of target mRNAs were engaged in cell growth and differentiation, nucleic acid binding transcription factor activity, and cancer-related pathways, according to an enrichment study of lncRNAs. Moreover, most the DEmiRNA-targeted mRNAs were engaged in immune system functions, signal transduction, and cancer-related pathways.

In addition, some studies have shown that miR-183 and miR-17 families can enhance the proliferation of bovine GCs [25,26]; miR-27a can promote the proliferation of mouse GCs [27], and a wide number of other miRNAs participate in promoting the proliferation of GCs. Nonetheless, a number of studies have also revealed that miRNAs hinder the proliferation of GCs. Zhao et al. discovered that miR-143-3p targets *BMPR1A* to induce apoptosis in human GCs [28], whereas Zhu et al. discovered that miR-222 suppresses apoptosis in porcine GCs via targeting the *THBS1* gene [29]. The primary explanation for these seemingly contradicting results is that follicular growth is dynamic, as are granulosa cell proliferation and apoptosis, and it is unbalanced for researchers to examine the entire process from a singular perspective.

Comparing the available high-throughput results in large and small follicles from Chuanzhong black goats in our laboratory [14], we discovered considerable enrichment for miR-450-5p and miR-202-5p. MiR-202-5p knockout reduced the expression of *CYP17, WNT2bb, WNT4a*, etc., retarding early follicular growth in medaka juveniles [10]. Studies of miR-202-5p in mammals have primarily focused on male gonads, while studies on the function of female gonads have been seldom reported [30,31]. Sequencing results revealed that the expression of miR-202-5p was 15 times higher in large follicles than in small follicles and that miR-202-5p was the most abundantly expressed DEmiRNA. It has been discovered targeting *TGFβR2* can trigger apoptosis in goat granulosa cells [11]. The primordial follicle develops to a point where the GCs are in a state of general proliferation, the GCs layer thickens, and the follicle volume dramatically increases. When the follicle is ovulated, the volume varies slowly, the follicle is filled with fluid, and the GCs are dominated by apoptosis [32,33]. Therefore, the mechanisms that regulate the proliferation and death of GCs during follicle development warrant further study. We discovered that miR-450-5p expression was elevated in big follicles and that high levels of miR-450-5p might target *EGFR* to regulate autophagy in order to increase glioma treatment biosensitivity [34]. The suppression of miR-450a-5p affects exosome-like vesicle-mediated adipogenesis in mice, according to a separate study [35]. In conclusion, it has been demonstrated that miR-450-5p is involved in cellular autophagy and adipose differentiation, but its role in mammalian folliculogenesis is rarely reported.

Apoptosis is essential for regulating proper mammalian development and maintaining the integrity of internal environment, and studies have demonstrated that the stimulation of PI3K/Akt signaling pathway decreases apoptosis in human trophoblast cells [36]. Wang et al. demonstrated that the AMPK-mTOR pathway promotes β-cell apoptosis [37], in addition to the MAPK signaling pathway, WNT signaling pathway, and TGF signaling pathway, which have all been established to influence cell proliferation and apoptosis [38,39,40]. Members of the *BCL2* family serve a crucial part in the apoptotic process, which are highly homologous and have preserved structural domains like BH1, BH2, BH3, BH4, etc. The *BCL2* family can be separated into two primary categories: anti-apoptotic proteins such as *BCL2, BCL-XL,* etc., and pro-cell death proteins such as *BAX, BMF*, etc. [41,42,43]. We screened the *BCL2* family to more precisely identify miR-450-5p and miR-202-5p target genes related to proliferation or apoptosis. Fortunately, we discovered *BMF* as a miR-450-5p target gene. A study has shown that knocking down *BMF* in breast cell lines prevented apoptosis and activated the MAPK pathway [44]. Notably, in a similar study, it was discovered that GCs undergo apoptosis. Li et al. discovered that *BMF* knockdown dramatically boosted the proliferation rate of goat GCs, while *BMF* overexpression increased GCs apoptotic rate [15] In the current study, we screened and confirmed *BMF* as a target gene of miR-450-5p and found that miR-450-5p targeting *BMF* activated the PI3K/Akt signaling pathway in GCs to regulate apoptosis. In addition, we identified *BCL2* as a target gene of miR-202-5p in goats, and the anti-apoptotic function of *BCL2* has been widely described. Moreover, we discovered that miR-202-5p impacts the AMPK signaling pathway to increase GCs death. In conclusion, we revealed that miR-450-5p increases cell proliferation and miR-202-5p promotes apoptosis, and both miRNAs influence GC proliferation synergistically, affecting follicle growth. Unfortunately, when separating large and small follicles, we had to sacrifice many medium follicles, which resulted in missing data for medium follicles in our results. Furthermore, for the following experimental validation, we only employed GCs from large follicles because we did not use magnetic bead sorting to separate GCs from tiny follicles, which resulted in a substantial number of theca cells intermingled with GCs from small follicles. These may lead to some limitations in our results. We also tried to find relevant DElncRNAs targeting mRNAs for further analysis of their functions. Although, lncRNAs annotation was incomplete due to the imperfect goat database, and few target mRNAs were predicted. However, the expression characteristics of DElncRNAs lead us to believe that they can be involved in follicle development with ineffable functions. And in the next work, more studies on the characteristics and functions of these lncRNAs will be conducted.

## 4. Materials and Methods

### 4.1. Ethics Statement

The experimental design was accepted by the Ethics Committee on Laboratory Animal Protection and Utilization of South China Agricultural University (license number: SYXK-2018-0123). In addition, all tests were conducted following the related rules of South China Agricultural University.

### 4.2. Laboratory Animals and Sample Collection

In this work, we utilized Leizhou goats grown on an experimental farm (College of Animal Science, South China Agricultural University, Guangzhou, China). To induce estrus in the female goats, 0.1 mg cloprostenol was injected [45].The goats were observed to determine whether or not they were in heat (bleating, searching for the male goat, frequent urination, hyperemia, edema, contraction of the vulva, and vaginal mucus discharge). The onset of estrus was characterized by the female goat shaking its tail, standing, and accepting mating with the male goats. The goats were butchered 24–36 h after confirming the estrus.

Ovaries were extracted from the ovarian bursa and promptly cleaned three times with 75% ethanol. They were then soaked in PBS. As reported previously, similar mechanical techniques were utilized to isolate ovarian follicles [46,47]. The follicles were created in 30 min using microblades and tweezers under a medical dissecting microscope. To generate RNA libraries, the isolated ovarian follicles were washed with PBS to remove debris, then immediately frozen in liquid nitrogen, and kept at −80 °C. Small follicles (SF, ≤3 mm in diameter) and large follicles (LF, ≥10 mm in diameter) were distinguished, and for the large follicles, the vascular-filled follicle was chosen, which indicates that the follicle is mature and ovulation-friendly. Only large follicles with clear follicular fluid and numerous blood vessels on their surface ovulate and are used for analysis. During follicular separation, follicles surrounding the large follicles are sacrificed, including small and medium-sized follicles (3 < d < 10 mm), and sometimes neighboring large follicles as well. Finally, within 30 min, 8 to 15 small follicles were isolated from each goat, 1 to 2 large follicles were isolated from each goat, and in several goats, there was only one large follicle with clear follicular fluid and abundant blood vessels on the surface, so we had to use only one large follicle as a replica for analysis. For each replicate, eight to fifteen small follicles from each goat were pooled and combined into a small follicle pool, while only one large follicle was collected”. The ovaries were harvested within twenty-four hours following the onset of estrus.

### 4.3. Library Preparation and High-throughput Sequencing

Total RNA was extracted from the follicles with Invitrogen TRIzol (Invitrogen, Carlsbad, CA, USA) according to the manufacturer’s protocol. 1% agarose gel electrophoresis was used to check the integrity of RNA, and the RNA integrity number (RIN) was evaluated by Agilent 2100 Bioanalyzer (Agilent Technologies, Palo Alto, CA, USA) [48]. RNA with an amount >6μg, concentration ≥200 ng/μL, 1.8 < OD_260/280_ < 2.2, and RIN > 8.5 was used to prepare cDNA library construction. Ribosomal RNA was removed from total RNA using Ribo-Zero™ Magnetic Gold Kit (EpiCentre, San Diego, CA, USA). A total of 3 μg RNA per sample was used to construct the LncRNA and mRNA library by Truseq TM RNA sample prep kit (Illumina, San Diego, CA, USA). The adapters were ligated to the 3’ and 5’ ends of the RNA, which had been fragmented into 200–300 base pairs. RNA fragments were reverse-transcribed into the first cDNA strand, and the second cDNA strand was specifically synthesized with the dUTP. Then, according to the effective concentration and the amount of data required for the library, the libraries containing different index sequences are mixed proportionally and uniformly diluted to form a single chain library through alkali denaturation.

The NEB Next Multiplex Small RNA library preparation kit was used to build the miRNA library using two micrograms of total RNA per sample (NEB, Beijing). After adaptor ligation, reverse transcription, and PCR, single-stranded cDNA was produced and utilized to generate a library of microRNA. The quantity of each library was determined using an Agilent 2100 Bioanalyzer, and its concentration was determined by qRT-PCR. Then, lncRNA and mRNA libraries were sequenced on the Illumina Hi Seq2500 platform (Illumina, San Diego, CA, USA) using the 150 bp Paired-End (BE) method with a depth of 25 M reads. The miRNA libraries were also sequenced using the Illumina Hi Seq2500 platform (Illumina, San Diego, CA, USA).

### 4.4. Data Processing

To obtain the clean reads, Cutadapt was adopted to filter the raw reads of FASTAQ according to the requirement of bioinformatics. The inferior quality sequence, such as adaptor sequences, overlap sequences, and sequences with quality scores <20, were removed. Q20, Q30, and GC content of the clean data were calculated (Appendix A). Then, the clean reads were aligned to the reference genome of Capra hircus (Accessed on 24 April 2019. http://asia.enSDbl.org/Capra_hircus/Info/Annotation) by Tophat2 after establishing the Genome index using Bowtie2 [49]. No more than two mismatches were regarded as successful mapping. StringTie was used to generate the transcripts according to the result of Tophat2 [50]. In particular, three softwares (CPC, CNCI, and Pfamscan) were used to predict the protein-coding potential of transcripts to screen out credible lncRNAs after generating the transcripts by StringTie. The clean reads of miRNA sequencing were mapped to the reference genome of Capra hircus using miRDeep2.

### 4.5. Differential Expression Analysis

In the transcriptome sequencing, the number of clean reads alignment to the reference genome was used to calculate the gene expression level. The statistics of the Read Count on each gene served as the original gene expression by HTSeq, and then FPKM was adopted to standardize the expression of mRNA and lncRNA (CPM in miRNA). Transcripts with the absolute values of log2foldchange > 1 and *p*-value < 0.05 were considered as significantly differentially expressed genes by DESeq. The visualization was also treated using the ggplots2 package of R.

### 4.6. Target Prediction

The cis prediction of lncRNA proposes that lncRNA placed upstream and downstream of protein-coding genes may control gene expression by interacting with promoters or other cis-acting elements of co-express genes. On the basis of the cis principle, the genes within 100 kb upstream or downstream of lncRNAs were served as close potential mRNA-lncRNA couples. For miRNA target prediction, the bioinformatics algorithms, including miRanda, miRDB, and Target scan7, were adopted to predict potential lncRNAs and mRNAs. Cytoscape (V3.9.1) was used to construct the DElncRNA-DEmiRNA-EmRNA interaction network between small and big follicles, and probable target pairs associated with differential expression were also found.

### 4.7. Functional Enrichment Analysis

Based on the candidate genes, functional annotation was implemented, including Gene Ontology (GO) and Kyoto Encyclopedia of Genes and Genomes (KEGG) enrichment analysis between small follicles and follicles. All DEmRNAs and target genes of lncRNA and miRNA were mapped to each term in Gene Ontology to determine the number of genes associated with each term. On the basis of differentially expressed (mRNA) genes and target genes, KEGG pathway analysis was performed (LncRNA). The criterion for statistical significance was FDR < 0.05.

### 4.8. CeRNA Network Analysis and Construction

The expression of mRNAs, miRNAs, and lncRNAs that differed significantly between small and large follicles was analyzed using ceRNAs. In addition, the mRNAs with low expression (FPKM < 1, and sample count > 3 in one group) will be filtered out and studied alongside DEmiRNAs and DElncRNAs. We searched the sequences of DElncRNAs and DEmRNAs for possible miRNA MREs. miRanda was used to predict miRNA binding sites, and the target interaction was overlapped with identical miRNA binding sites in the lncRNA and mRNA sequences. On the basis of the aforementioned target relationships, the ceRNA network was created and visualized using the Cytoscape program (V3.9.1).

### 4.9. Cell Culture

For this experiment, the used KGN (ovarian granulosa cells) and 293T (human embryonic kidney cells) cells were kept. KGN and 293T cells were grown in DMEM/F-12 medium DMEM culture media (Invitrogen, Carlsbad, CA, USA), respectively. 10% fetal bovine serum (FBS, Fisher, NY, USA), 100 units/mL penicillin, and 100 g/mL streptomycin were added to the medium. Then they were placed in an incubator at 37 °C., 5% CO2, and 95% humidity (Invitrogen, Carlsbad, CA, USA). Depending on the goal of this experiment, 10cm plates, 6-well plates, and 96-well plates (Invitrogen, Carlsbad, CA, USA) were utilized.

### 4.10. Transfection

MiRNA mimics, mimic NC, and siRNA sets were designed and synthesized by Ribo (Guangzhou, China). Synthetic sequence information is shown in Appendix A. Lipofectamine™ 3000 Transfection Reagent (Invitrogen, Carlsbad, CA, USA) and synthetic liposomes were added according to the manufacturer’s instructions. Pbs Equilibrium Buffer (Invitrogen, Carlsbad, CA, USA) was used to wash the cells, and Opti-MEM I Minus Serum Medium (Invitrogen, Carlsbad, CA, USA) was used to dilute the liposomes. The transfection efficiency was determined by qRT-PCR.

### 4.11. Cell Counting Kit-8

The Cell Counting Kit-8 was used to detect the proliferation of cells according to the kit instructions. Usually, cell proliferation experiments are performed by adding 100 µL of culture medium per well as well as inoculating 2000 cells using 96-well cell culture plates. Add 10 µL of CCK-8 solution to each well and continue incubation for 0.5–4 h in the cell culture incubator. Measure the absorbance at 450 nm. Six replicates of each group were used, and the mean value was calculated.

### 4.12. EdU Cell Proliferation

Following the instructions, experiments were performed using the BeyoClick™ Edu-488 Cell Proliferation Assay Kit (beyotime, Shanghai, China). The prepared EDU reagent was added to the cell culture medium and incubated for 2 h; after that, the medium was removed, and fixation solution was added for 10 min, followed by permeabilization solution for 10 min, after staining the nuclei.

### 4.13. Dual Luciferase Reporter Gene Assay

The dual fluorescent null plasmid PGLOmir was saved for this experiment, and the 3’ UTR of BMF gene was downloaded from the NCBI database, intercepted with the miR-450-5p target binding sequence and its upstream and downstream 200 bp, constructing into PGLOmiR to form a wild-type plasmid. Likewise, the binding sequence was all mutated to base C to form a mutant plasmid. This experiment was performed according to the instructions of Dual Luciferase Reporter Gene Assay Kit (Yeasen Biotechnology; Shanghai, China). Cells were transfected using 24-well plates. Then, 200 µl cell lysis solution was added and lysed on ice for 5 min, and the supernatant was taken for the experiment after centrifugation at 10,000 rpm for 1 min. The fluorescence detection procedure is as follows: First, 20 μL lysate was taken and added to the culture plate with six replicates according to the experimental needs. Then, 100 μL firefly fluorophore enzyme reaction solution was added, the plate was shaken and mixed, and the vitality of firefly fluorophore enzyme was detected. Finally, 100 μL sea kidney fluorophore enzyme reaction solution was added, the plate was shaken and mixed, and the vitality of sea kidney fluorophore enzyme was detected, and the whole assay was completed within 30 min as much as possible.

### 4.14. Western Blot

After the transfected cells were washed twice in 6-well plates with PBS, 150 μL RIPA III lysis solution (Sangon Biotech, Shanghai, China) was added and lysed on ice for 15 min. The cells were centrifuged at 4 °C for 2 min at 12,000× *g* rpm, and the supernatant was transferred to a new centrifuge tube. The protein concentration was determined by BCA method (Invitrogen, Carlsbad, CA, USA), and then the protein samples were boiled in 5 × SDS protein loading buffer (Solarbio, Beijing, China) at 100 °C for 10 min. Two micrograms protein from each cell extract were resolved by 10% Acr-Bis SDS-PAGE and transferred to polyvinylidene difluoride membranes (PVDF, Millipore). Blots were detected by incubating with the antibodies BMF (Sangon Biotech, NO. D162765, 1:1000 dilution), P-AKT (Cst, #4060S, 1:1000 dilution), AKT (Cst, #4685S, 1:1000 dilution), BCL2 (Cst, #4223, 1:1000 dilution), P-AMPK (Cst, #4184, 1:1000 dilution), AMPK (Cst, #5831, 1:1000 dilution) or β-Actin at 4 °C overnight. Immunoreactive bands were then probed with appropriate horseradish peroxidase (HRP)-coupled secondary antibodies, such as goat anti-rabbit Ig G-HRP or goat anti-mouse Ig G (H + L)/HRP at room temperature for 2 h, and then protein bands were detected with chemiluminescent HRP substrate (Invitrogen, Carlsbad, CA, USA).

### 4.15. Detection of Ovarian Granulosa Cell Apoptosis by Flow Cytometry

Experiments were performed according to the Annexin V-FITC/PI Apoptosis Detection Kit instructions. First, cells were collected after digestion with EDTA-free trypsin by. Cells were washed twice with pre-chilled PBS, followed by centrifugation at 4 °C, 300× *g* for 5 min. 1~5 × 10^5^ cells were collected. Aspirate PBS and add 100 μL 1×Binding Buffer to resuspend the cells. Add 10 μL Annexin V-FITC and 10 μL LPI Staining Solution and mix gently. Add 400 μL of 1 × Binding Buffer, mix well and place on ice. Samples were detected within 1 h by BD FACSAria™ Fusion Flow Cytometer (BD, Franklin, NY, USA).

### 4.16. Real-Time qPCR Validation 

To validate the data from RNA sequencing, a total of 10 DEmRNAs, 5 DELncRNAs, and 2 DEmiRNAs were selected to be validated with quantitative real-time PCR (qRT-PCR). The samples of qRT-PCR came from the RNA remaining from sequencing. For the mRNA and lncRNA validation, the total RNA was reverse transcribed into complementary DNA (cDNA) using PrimeScript RT Reagent Kit With gDNA Eraser (Takala, Beijing), and the specific quantitative primers of mRNA and lncRNA are listed in Appendix A. The Real-Time PCR was performed on QuantStudio 7 Flex Real-Time PCR System (Thermo Fisher Scientific, Carlsbad, CA, USA) with the following protocol: 95 °C for 1 min, followed by 35 cycles of 95 °C for 30 s, 60° C for 30 s, and 72 °C for 1 min with the PowerUp SYBR Green Master Mix (Thermo Fisher Scientific, Carlsbad, CA, USA). For the validation of miRNA, the total RNA was reverse transcribed into complementary DNA (cDNA) after adding Poly A using miDETECT A TrackTM miRNA qRT-PCR Starter Kit (Ribobio, Guangzhou, China). The primers were synthesized by Ribobio (Guangzhou, China). The Real-time PCR was performed on CFX96 Touch Deep Well Real-Time PCR Detection System (Bio-Rad, Hercules, CA, USA) with the following protocol: 95 °C for 10 min, followed by 40 cycles of 95 °C for 2 s, 60 °C for 20 s, and 70 °C for 10 s with the miDETECT A TrackTM miRNA qRT-PCR Starter Kit (Ribobio, Guangzhou, China). Primer information was listed in Appendix A.

### 4.17. Statistical Analysis

Comparisons between two groups or multiple groups were performed using *t*-test and one-way variance analysis, respectively. All data were expressed as the mean ± standard deviation (SD), with at least three independent replicates. GraphPad Prism (version 9.0) was used for all analyses (GraphPad Software, La Jolla, CA, USA). The significance thresholds for differences were *p* < 0.05 for *, *p* < 0.01 for **, and *p* < 0.001 for ***.

## 5. Conclusions

In conclusion, we provided a transcriptome study of DEmRNAs, DElncRNAs, and DEmiRNAs in small and big goat follicles. MiR-450-5p influences PI3K/Akt signaling to enhance GCs proliferation, whereas miR-202-5p limits GCs proliferation by impacting the AMPK signaling. Importantly, we determined that *BMF* and *BCL2* are the target genes of miR-450-5p and miR-202-5p in goats, respectively. This study offers the groundwork for a comprehensive examination of the activities of lncRNA-miRNA-mRNAs that play an important role in goat follicle development.

## Figures and Tables

**Figure 1 ijms-24-00401-f001:**
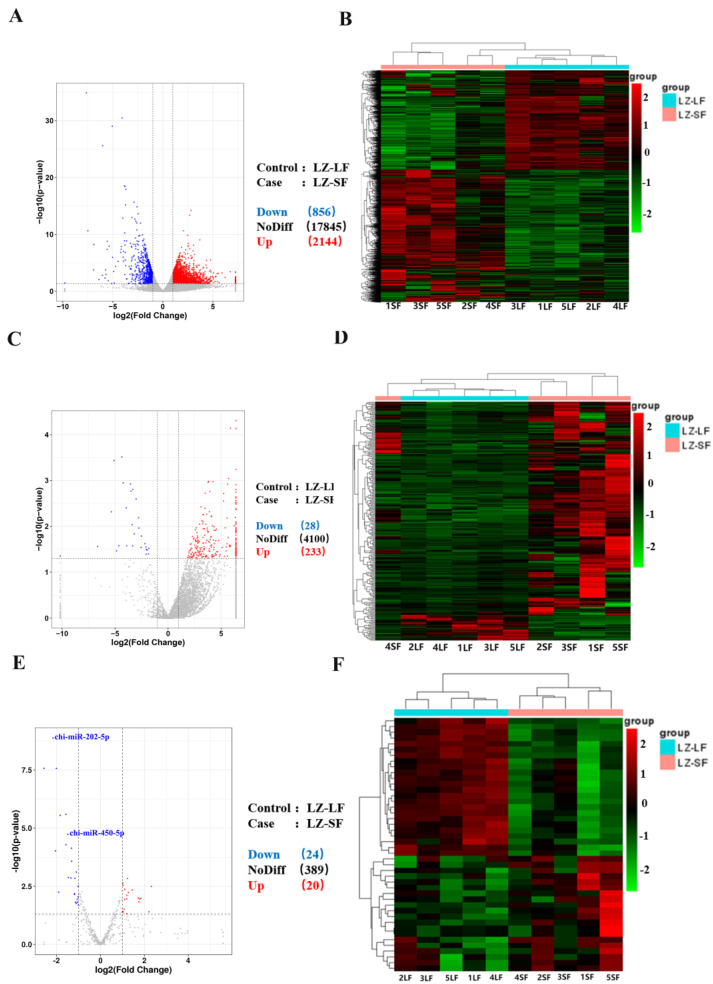
Volcano map and cluster analysis of DEmRNAs, DElncRNAs, and DEmiRNAs. Volcano plots of DEmRNAs (**A**), DElncRNAs (**C**) and DEmiRNAs (**E**); Cluster analysis plots of DEmRNAs (**B**), DElncRNAs (**D**) and DEmiRNAs (**F**).

**Figure 2 ijms-24-00401-f002:**
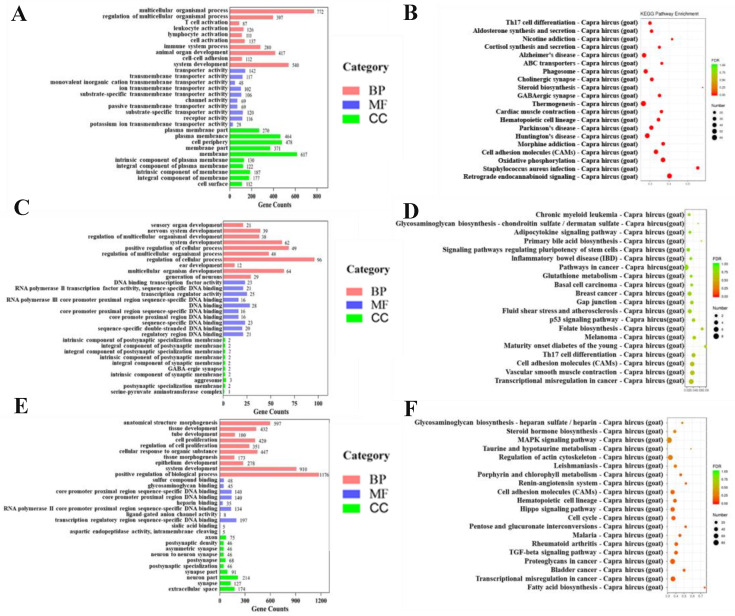
GO and KEGG enrichment analysis of DEmRNAs and target genes of DElncRNAs and DEmiRNAs. GO analysis of DEmRNAs (**A**) and predicted target genes of DElncRNAs (**C**) and DEmiRNAs (**E**). The top 20 KEGG pathways of predicted target genes for DEmRNAs (**B**) and DElncRNAs (**D**), and DEmiRNAs (**F**). “Count” means the number of predicted target genes enriched in this pathway. The color represents the degree of enrichment, with red representing significant enrichment. Red represents the biological process (BP), blue represents the molecular function (MF), and green represents the cell component (CC).

**Figure 3 ijms-24-00401-f003:**
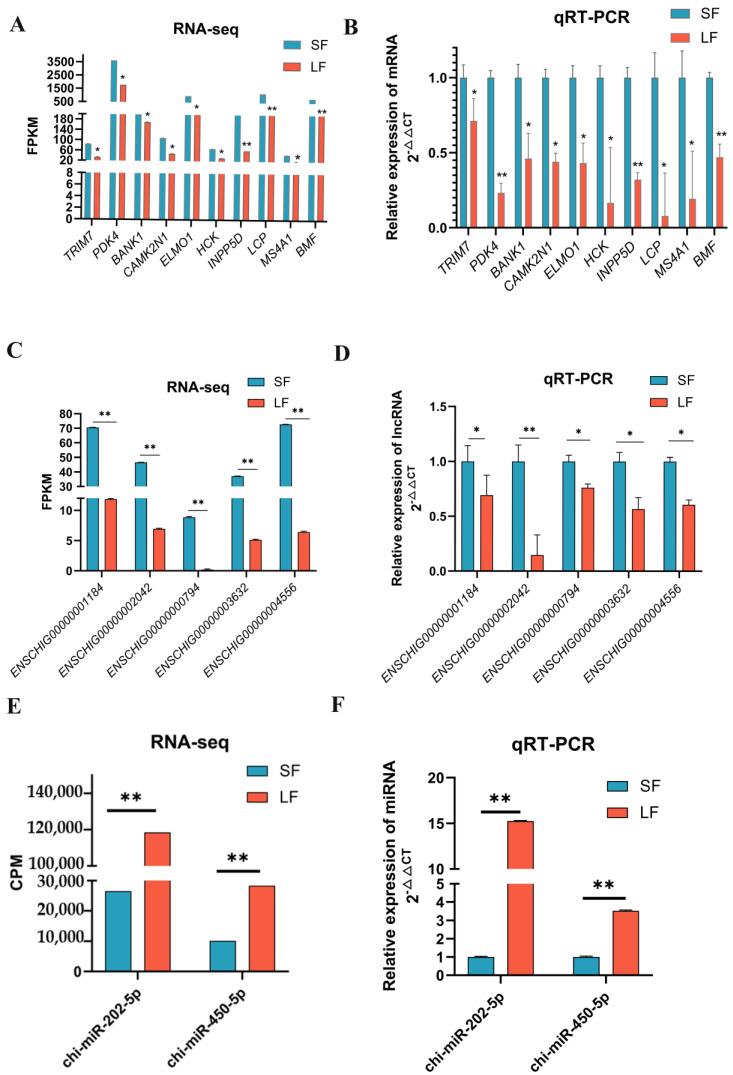
Differentially expressed mRNAs, lncRNAs and miRNAs were verified by qRT-PCR. (**A**,**B**): The expression levels of the 10 mRNAs were verified by qRT-PCR and compared with the results of RNA-seq. (**C**,**D**): The expression levels of the five lncRNAs were verified by qRT-PCR and compared with the results of RNA-seq. (**E**,**F**): The expression levels of the 2 miRNAs were verified by qRT-PCR and compared with the results of RNA-seq. The results were shown to be consistent with RNA-seq data by RT-qPCR. Data are shown as mean ±SD values (*n* = 3, * *p* < 0.05; ** *p* < 0.01 indicates significant difference).

**Figure 4 ijms-24-00401-f004:**
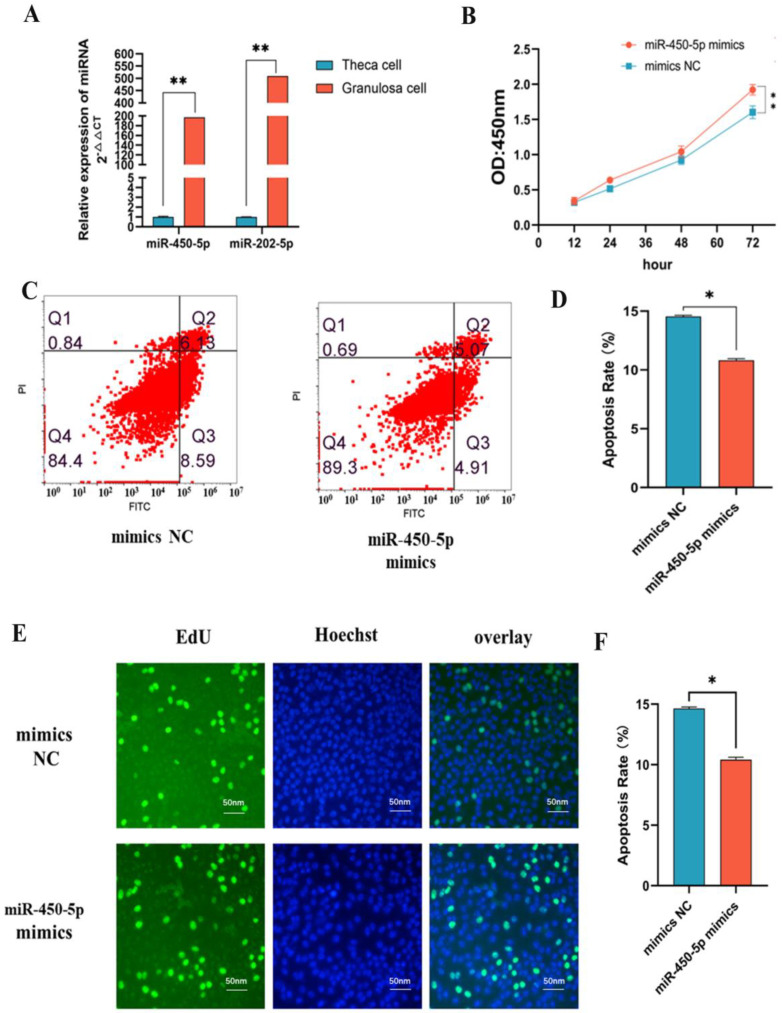
miR-450-5p stimulates the proliferation of GCs. (**A**) Expression of miRNA in follicular granulosa cells versus follicular theca cells. Results are shown as Mean ± SD, *n* = 4, ** *p* < 0.01 indicates a highly significant difference. (**B**) CCK8 line graph of miR-450-5p mimics and mimics NC. OD values at 450 nm were determined at 12 h, 24 h, 48 h, and 72 h. (Mean ± SD, ** *p* < 0.01). (**C**,**D**) Results of flow apoptosis measurements of miR-450-5p mimics versus mimic NC with apoptotic cells as a percentage of all cells. (Mean results ± SD, * *p* < 0.05 indicates significant difference). (**E**,**F**) EdU cell proliferation assay. Comparison of miR-450-5p mimics as well as mimics NC in EdU fuel (proliferating cells) as a percentage (%) of hoechst fuel (overall cells). Three replicates per group, mean ± SD,* *p* < 0.05 indicates a significant difference.

**Figure 5 ijms-24-00401-f005:**
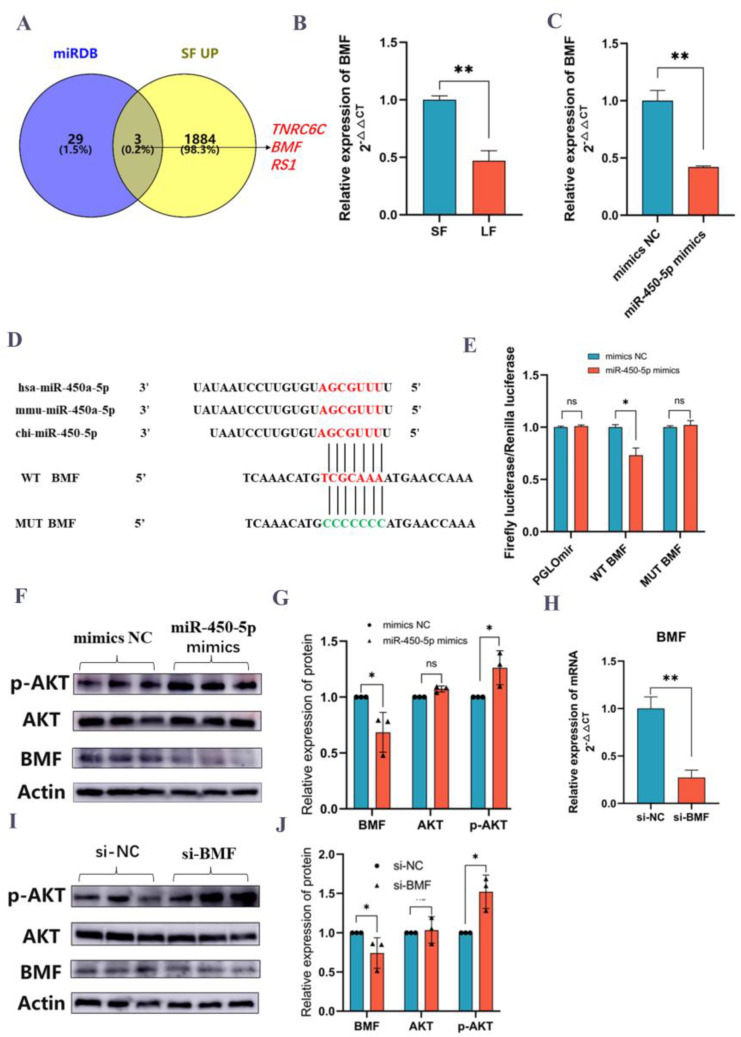
miR-450-5p targets *BMF* to activate PI3K/Akt signaling pathway (**A**): Comparison of genes upregulated in small follicles with target genes of miR-450-5p predicted by miRDB database. (**B**) *BMF* gene expression on small and large follicles in goats. (**C**) Expression of *BMF* gene after transfection with miR-450-5p. (**D**) Sequence comparison of miR-450-5p and comparison of WT-BMF and MUT-BMF target sequences. (**E**) Dual fluorescence changes of the original plasmid pGLOmir, wild plasmid WT, and mutant plasmid MUT with miR-450-5p (**F**,**G**) Western blot results of p-AKT, AKT, and *BMF* after transfection of miR-450-5p. (**H**) Expression of *BMF* gene after silencing *BMF* with siRNA. (**I**,**J**) Western blot results of p-AKT, AKT, and *BMF* silenced with siRNA. Reported experiments. Data are shown as mean ± SD values (*n* = 3). * *p* < 0.05; ** *p* < 0.01was considered statistically significant. ns: not significant for statistical results.

**Figure 6 ijms-24-00401-f006:**
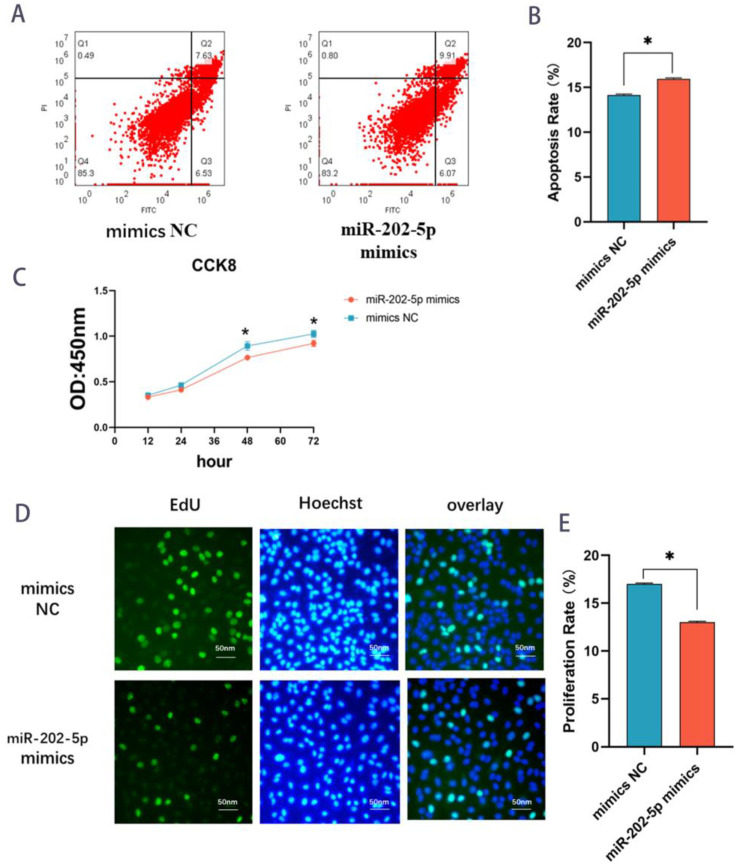
miR-202-5p induces the apoptosis of follicular GCs. (**A**,**B**) Flow cytometric results of miR-202-5p mimics vs. mimics NC. Flow cytometric results histogram of miR-202-5p mimics vs. mimics NC. (Mean results ± SD, * *p* < 0.05 indicates significant difference) (**C**) CCK8 line graph of miR-202-5p mimics and mimics NC. OD values at 450 nm were determined at 12 h, 24 h, 48 h, and 72 h. (Mean ± SD, * *p* < 0.05). (**D**,**E**) EdU cell proliferation assay. Comparison of miR-202-5p mimics as well as mimics NC in EdU fuel (proliferating cells) as a percentage (%) of hoechst fuel (overall cells). Three replicates per group, mean ± SD, * *p* < 0.05 indicates a significant difference.

**Figure 7 ijms-24-00401-f007:**
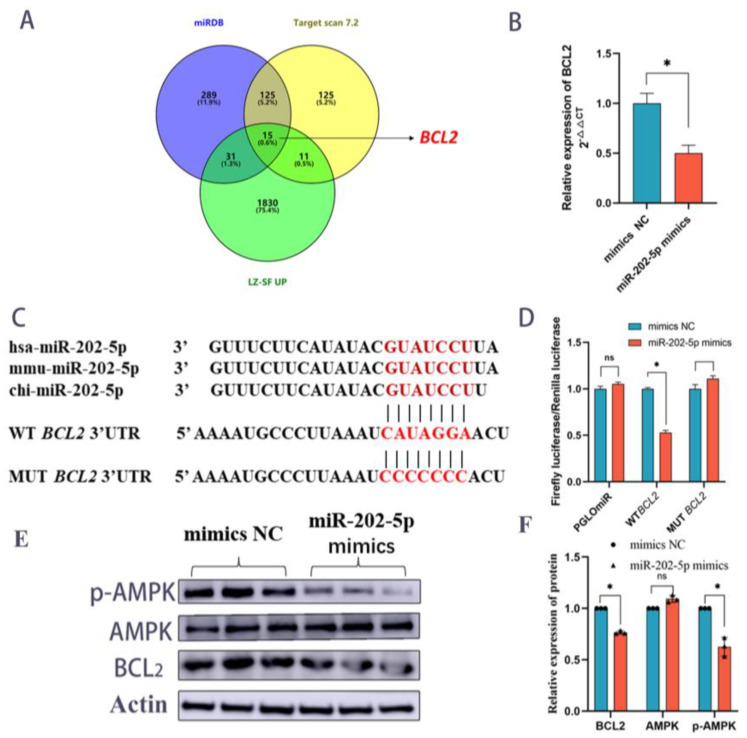
miR-202-5p targets the *BCL2* gene. (**A**) miRDB with Target scan 7.2 databases to predict target genes. (**B**) qPCR results of *BCL2* gene after overexpression of miR-202-5p. (**C**) Sequence comparison of miR-202-5p with dual luciferase results. (**D**) Fluorescence changes of miR-202-5p and mimics NC after co-staining with PGLOmir, WT-BCL2 and MUT-BCL2. Data are shown as mean ± SD values (*n* = 6). * *p* < 0.05 was considered statistically significant. Ns: data results were not significant. (**E**,**F**) Western blotting results of p-AMPK, AMPK and BCL2 after overexpression of miR-202-5p.Data are shown as mean ± SD values (*n* = 3). * *p* < 0.05 was considered statistically significant. ns: not significant for statistical results.

**Figure 8 ijms-24-00401-f008:**
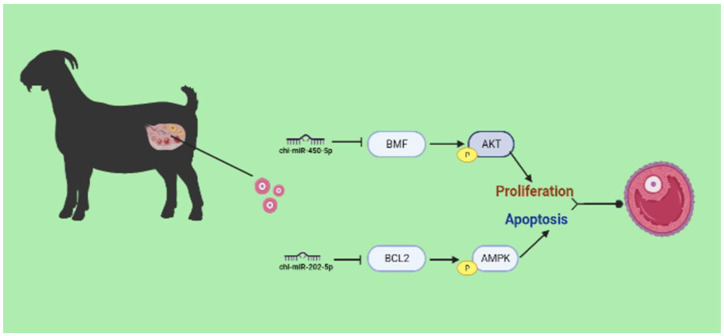
Diagram of miR-450-5p and miR-202-5p regulatory follicle development pathway.

**Table 1 ijms-24-00401-t001:** Quality control of RNA-seq data.

Sample	Clean Data (bp)	Clean Reads (%)	Q30 (bp)	Q30 (%)
1LF	15585066260	99.59	14501525545	92.22
2LF	15140200466	99.69	14268818395	93.70
3LF	15163799328	99.76	14323828383	94.01
4LF	15157744912	99.53	14396101261	94.29
5LF	15107316988	99.62	14075908419	92.57
1SF	15770894688	99.27	14633479434	91.80
2SF	15601313394	99.75	14731089790	93.89
3SF	15205292826	99.34	14270121732	92.95
4SF	15779273590	99.67	14771329420	93.05
5SF	15831519850	99.65	14829551313	92.90

Q30 value represents the error probability of 0.1% for the identified bases in the process of base recognition;1LF-5LF, large follicles;1SF-5SF, small follicles.

**Table 2 ijms-24-00401-t002:** The top ten DEmRNAs, DElncRNAs, and DEmiRNAs were found to be significantly different between large and small follicles.

DE	log2FoldChange	*p* adj.
DEmRNAs		
SLC24A5	−7.63797175	2.75 × 10^−31^
INHBA	−4.070975591	3.51 × 10^−27^
INHA	−5.053598506	6.63 × 10^−26^
REG4	−6.02283724	1.31 × 10^−22^
TMEM200A	−3.819041739	1.27 × 10^−15^
SYCN	−3.778589892	1.28 × 10^−15^
ESM1	−3.690845736	3.66 × 10^−15^
TMEM132A	−2.885813128	5.69 × 10^−13^
TFR2	−3.722397945	1.51 × 10^−12^
MFGE8	−2.581577809	2.88 × 10^−12^
DElncRNAs		
ENSCHIT00000001444	6.403989	0.0135566
MSTRG.4906.1	5.88433284	0.0182051
ENSCHIT00000005590	Inf	0.0183993
ENSCHIT00000005038	−4.361957153	0.0506088
MSTRG.7728.1	−5.091859075	0.0571997
ENSCHIT00000011140	Inf	0.0769421
ENSCHIT00000001013	5.710455636	0.1042035
MSTRG.16601.1	3.827631701	0.1155763
ENSCHIT00000001457	4.242619114	0.1162408
ENSCHIT00000002724	3.775845698	0.1187464
DEmiRNAs		
chi-miR-202-5p	−2.160010803	5.71 × 10^−7^
chi-miR-202-3p	−2.574623058	3.93 × 10^−6^
chi-miR-190b	−2.005967281	3.93 × 10^−6^
chi-miR-144-5p	−1.836108259	0.0002434
chi-miR-451-5p	−1.578914668	0.0002434
chi-miR-450-5p	−1.486338173	0.0013283
chi-miR-144-3p	−1.574316038	0.0031894
chi-miR-128-3p	−1.320891702	0.0039405
chi-miR-424-3p	−2.049938746	0.0045749
chi-miR-409-5p	−1.314150135	0.0115349

**Table 3 ijms-24-00401-t003:** Sequencing differential expression table.

id	Gene Name	log2FoldChange	*p* adj.	Regulation
ENSCHIG00000017926	BMF	1.826718012	0.006168	UP
ENSCHIG00000026004	TNRC6C	1.054662056	0.020518	UP
ENSCHIG00000022783	RS1	Inf	0.036202	UP
ENSCHIG00000009035	miR-450-5p	−1.486338173	0.001328	DOWN

## Data Availability

The data presented in this study are available on request from the corresponding author.

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
