# Peer review of "miR-450-5p and miR-202-5p Synergistically Regulate Follicle Development in Black Goat"

_ijms, 2022, doi:10.3390/ijms24010401_

Round 1
Reviewer 1 Report
Manuscript title: miR-450-5p and miR-202-5p synergistically regulate follicle development in black goat
Authors: Feng G et l.,
Manuscript ID: ijms-2040207
Interesting study. The authors investigated regulatory mechanisms of lncRNAs,
mRNAs, and microRNAs, RNA sequencing (RNA-seq) and small RNA-seq were performed on
large follicles (>10 mm) and small follicles (<3 mm) of Leizhou black sheep during estrus.
- Though the authors provided time of follicle collection after estrus, they needed to mention what phase (recruitment, maturation or atretic) of follicular wave the follicles (large and small) were collected. Also, they need to indicate whether both large and small follicles collected from each replicates. These needed to be clarified.
- For eg: 10 mm follicle are found at maturation or atretic phase. This may influence type of genes expressed and involved. Similarly, the small follicle may be suppressed by the presence of large follicles.
- Considering the following two points the question is whether authors considered number of granulosa cells (count) in RT-PCR methods to determine expression levels?
1. These results imply that during follicular growth in black goats, the miR- 32 450-5p-BMF axis favored GC proliferation on a wide scale, while the miR-202-5p-BCL2 axis triggered GC apoptosis.
2. Figure 2 qRT-PCR result shows both miRNA expression in small and large follicles follows similar pattern.
- Section 2.7 and 2.9: Clarify whether GCs from large or small follicles in this experiment? If only one GC from one follicle type is used clarify why other type of CGs were not used.
- Line 192: The author claims “The current study discovered that miR-450-5p was highly expressed in large follicles”. I did not notice such comparison in this version. Please include the evidences for such comparison.
- The conclusions needed to be carefully drafter after there clarifications.
Reviewer 2 Report
Regarding the manuscript entitled “miR-450-5p and miR-202-5p synergistically regulate follicle development in black goat”, the authors aimed to investigate the regulatory mechanisms of lncRNAs, mRNAs, and microRNAs. RNA sequencing (RNA-seq) and small RNA-seq were performed on large follicles (>10 mm) and small follicles (<3 mm) of animals during estrus.
The study is interesting. However, major revisions and English editing are needed before its acceptance.
The authors wrote “Leizhou black sheep” in line 21. While in title and material and methods, they wrote black goat!!!!!!!!!!!!!!!!
Were animals goat or sheep?!!!!!!!!!
What did the authors do with follicles 3-10mm in diameter?
Why the authors did not use eCG to get more follicles?!
“while only one giant follicle was pooled” pooled with what?!
Round 2
Reviewer 1 Report
These explanations provided for all 6 points needed to be elaborated in the M&M and discussion sections. The discussion should also include limitations.
For example one of the explanation provided is "Despite the fact that blunt separation still faces the issue of impure cell separation and contamination, the number of GCs from large follicles is sufficient for us to eliminate those that do not meet the requirements."
There are methods available to avoid this type of contamination while cell separations by using cell sorting by magnetic beads.
Reviewer 2 Report
The authors improved their manuscript following the reviewers' recommendations.
Author Response
Dear Editor:
Thank you for a rapid processing of our manuscript exclusively submitted for publication in International Journal of Molecular Sciences. We hereby submit a revised version of the manuscript “miR-450-5p and miR-202-5p synergistically regulate follicle development in black goat” (Manuscript ID: ijms-2040207`). We thank the reviewers for their constructive criticism that has helped us to improve the manuscript. We have carefully revised our manuscript, hoping that our work could be accepted for publication.
Yours Sincerely,
Guanghang Feng
On behalf of all co-authors
Email: fgh@stu.scau.edu.cn